# Risk factors for malaria-related mortality among children under five at Mbale Regional Referral Hospital, Uganda, 2020–2024: A case-control study

Patrick Kwizera[1]*, Richard Migisha[1], Charity Mutesi[1], Gerald Rukundo[2], Steven Ndugwa Kabwama[1], Benon Kwesiga[1], Lilian Bulage[1], Alex Riolexus Ario[1]

1 Uganda Public Health Fellowship Program, Uganda National Institute of Public Health, Kampala, Uganda, 2 National Malaria Elimination Division, Ministry of Health, Kampala, Uganda

* pkwizera@uniph.go.ug

## Abstract

Malaria remains a major global health burden, with 264 million cases and 569,000 deaths in 2023. Uganda ranks third globally in malaria cases and tenth in deaths, with 95% of the country endemic and children under five most vulnerable. Despite control efforts, Mbale Regional Referral Hospital (MRRH) in Eastern Uganda reported a pediatric malaria case fatality rate of 2.7% between 2020 and 2024. This study aimed to identify factors associated with malaria-related deaths among children under five admitted to the hospital during this period. We conducted an unmatched 1:2 case–control study using retrospective data from 2020–2024. Cases were children ≤59 months with WHO-defined severe *Plasmodium falciparum* malaria who died during hospitalization (n = 100). Controls were similar children who recovered (n = 200), systematically sampled from about 32,400 admissions. Data were extracted from patient records, and multivariable logistic regression identified mortality predictors. Among 100 cases, 73% were aged <24 months and 61% were male. Convulsions (adjusted odds ratio [aOR]=17; 95%CI:4.2–71), loss of consciousness (aOR=14; 95%CI: 1.4–113), severe anemia (aOR=3.4; 95%CI:1.4–8.2), vomiting (aOR=3.1; 95%CI: 1.4–6.9), and delays in seeking care > 24 hours after symptom onset (aOR=8.8; 95%CI: 2.3–34) were significantly associated with mortality.Malaria deaths among under-five children was significantly associated with severe clinical features,convulsions, loss of consciousness, and severe anemia and delayed care-seeking. Early recognition of danger signs, and prompt care-seeking could reduce paediatric malaria mortality in high-burden settings like Mbale.

**Data availability statement:** The datasets upon which our findings are based belong to the Uganda Public Health Fellowship Program. For confidentiality reasons, the datasets are not publicly available. The datasets can be availed upon reasonable request from the responsible officer with permission from the Uganda Public Health Fellowship Program. Request can be directed at: it@uniph.go.ug.

**Funding:** This study was supported by the President's Emergency Plan for AIDS Relief (PEPFAR) through the United States Centers for Disease Control and Prevention Cooperative Agreement number GH001353-01 to PK, RM, CM, GR, SNK, BK, LB, AR through Makerere University School of Public Health to the Uganda Public Health Fellowship Program, Ministry of Health. The funders had no role in study design, data collection and analysis, decision to publish, or preparation of the manuscript.

**Competing interests:** The authors have declared that no competing interests exist.

## Background

Malaria is one of the most significant global health challenges. In 2023 alone, there were 264 million cases and 569,000 death, with the World Health Organization (WHO) African Region bearing a heavy burden [1]. Between 2019–2023, both the number of malaria cases and deaths increased, rising by 23 million and 24,000 respectively [1]. Globally, Uganda ranks as the third highest contributor to malaria cases and the tenth highest contributor to malaria-related deaths, according to the World Malaria Report (WMR) 2024 [1, 2].

As of 2022, malaria was endemic in approximately 95% of Uganda, with the remaining 5% of the country being susceptible to malaria epidemics [3, 4]. The disease accounts for 30–50% of outpatient visits, 15–20% of hospital admissions and up to 20% of inpatient deaths [5–7]. Malaria-related mortality continues to challenge Uganda's healthcare system, especially in rural and high-burden districts where access to timely diagnosis and treatment may be limited [8].

Children <5 years of age bear the highest burden of malaria, as they are most vulnerable to severe forms of the disease and its complications. The World Health Organization (WHO) reports that this age group accounts for a significant proportion of global malaria deaths, many of which are preventable with early diagnosis and effective treatment [1].

Malaria is a mosquito-borne infectious disease caused by protozoan parasites of the genus *Plasmodium*, transmitted through the bite of infected female *Anopheles* mosquitoes. The disease is characterized by the presence of *Plasmodium* parasites in the bloodstream, where they invade and multiply within red blood cells [9, 10]. In sub-Saharan Africa, *Plasmodium falciparum* is the predominant species responsible for most severe disease and malaria-related mortality [11, 12]. Severe malaria occurs when confirmed *P. falciparum* infection is accompanied by life-threatening clinical or laboratory evidence of vital organ dysfunction. According to WHO, this includes impaired consciousness or coma, severe anemia (hemoglobin <7 g/dL), acute kidney injury, respiratory distress, circulatory collapse or shock, metabolic acidosis, jaundice, or disseminated intravascular coagulation [13]. Without urgent intervention, these complications can lead to rapid deterioration and death, especially in young children.

In Uganda, children under five represent a large proportion of malaria-related admissions and deaths [14]. To address this ongoing challenge, Uganda has implemented several primary interventions for malaria control, including the distribution of insecticide-treated bed nets (ITNs) and prompt diagnosis and treatment with artemisinin-based combination therapy (ACT). Additionally, integrated community case management of childhood illnesses (iCCM) and indoor residual spraying (IRS) have been implemented in some high burden areas [2]. However, despite these efforts, malaria-related mortality remains high, particularly among children under five. Between 2020 and 2024, Mbale Regional Referral Hospital (MRRH) recorded 13,903 pediatric malaria cases, of which 376 resulted in death, yielding a mortality rate of 2.7%, substantially higher than the national average of 0.6% [15]. This persistently high rate suggests that additional factors such as delayed care seeking,

severe anemia and other clinical or health system-related determinants, may be influencing outcomes and limiting the effectiveness of existing interventions [16]. Therefore, this study aimed to identify the key factors contributing to severe malaria-related mortality at Mbale Regional Referral Hospital, Eastern Uganda, 2020–2024.

## Methods

### Study design and study area

We conducted an unmatched 1:2 case control study, utilizing routine surveillance data extracted from hospital records. Data were accessed between April 7th and 15th,2025. The study was conducted at Mbale Regional Referral Hospital located 250 km east of Kampala, in Mbale City. The hospital serves a catchment area of over 4.6 million people across 16 districts: Budaka, Bududa, Bukwo, Bulambuli, Busia, Butaleja, Butebo, Kapchorwa, Kibuku, Kween, Manafwa, Mbale, Pallisa, Namisindwa, Sironko, and Tororo. The hospital provides both primary and specialized care, playing a crucial role in the management of a wide range of medical conditions within the area. Malaria is diagnosed mainly using microscopy and malaria rapid diagnostic test (mRDT) from peripheral blood. Malaria is classified into two categories uncomplicated, and severe forms. Uncomplicated malaria is diagnosed using rapid diagnostic tests in health facilities but also in the community by the trained village health teams (VHTs). Severe malaria cases are referred to the hospital for further management where artesunate is used as first line treatment [2].

### Study population

The study population consisted of all children under 5 years who were admitted at Mbale Regional Referral Hospital from 1 January, 2020–31 December, 2024. Specifically, the focus was on children who died from malaria or related complications during this period.

### Case and control definitions

For the purposes of this study, we defined cases and controls as follows:
**Case:** A child aged ≤59 months who was admitted to Mbale Regional Referral Hospital from 1 January, 2020–31 December, 2024 with severe *Plasmodium falciparum* malaria, as defined by WHO criteria, who died during hospitalization.
**Control:** A child aged ≤59 months who was admitted to the same hospital during the same period with severe *Plasmodium falciparum* malaria (WHO-defined), who survived and was discharged from the hospital.

### Sample size determination and sampling procedure

The required sample size was estimated using the Fleiss formula with continuity correction, which is widely applied for calculating sample sizes in unmatched case–control studies comparing proportions of exposure between cases and controls [17]. The calculation assumed 80% power to detect an odds ratio of 2.0 at a 5% significance level, with an expected prevalence of severe anemia among controls of 26%, based on previous studies from Rwanda [16]. This yielded a target of 121 cases and 242 controls. However, since only 100 deaths due to severe malaria in children ≤59 months were identified at MRRH during 2020–2024, we included all these as cases and retained a 1:2 case-to-control ratio, resulting in 200 controls. The sample size was calculated using Epi Info version 7.2.6.0 (STATCAL module).

Controls were selected from 32,400 under-five children admitted with laboratory-confirmed malaria who survived during the same period. Systematic sampling was employed to select 200 controls: the sampling interval (k) was calculated by dividing the total eligible controls by the desired number of controls (k = 32,400 by 200 = 162); a random starting point between 1 and k was selected, and every kth record thereafter was included until 200 controls were sampled.

## Data collection

Data were collected using a standardized abstraction tool developed for this study and administered by two trained research assistants. After a one-day training session, the assistants retrieved patient files from physical archives. Initial demographic information such as age, sex, residence, and insurance status were obtained from hospital registers. This was followed by extraction of detailed clinical, case management, and healthcare access data from patient records, including symptoms at admission, malaria complications, treatments received (antimalarials, antibiotics, blood transfusion), duration of hospital stay, and, where applicable, timing and cause of death. Additional information on healthcare access (referral status, time to admission, mode of arrival), diagnostic testing, and environmental factors such as seasonality and admission shift was also collected. Data quality was maintained through daily reviews and cross-checking with available records.

## Inclusion and exclusion criteria

Children aged 59 months and below with a confirmed positive malaria test (microscopy or mRDT) and presenting with severe symptoms were included in the study. We excluded children who were eligible but lacked essential information on symptoms, treatment, or outcomes, as accurate data extraction and analysis was not possible for these records.

## Data analysis procedures

Descriptive statistics were used to summarize sociodemographic, clinical, and case management characteristics. Categorical variables were presented as frequencies and percentages. The Chi-squared test was used to assess associations between independent variables and the outcome variable. Logistic regression analysis was performed to identify factors associated with severe malaria-related mortality among children. Significant variables at bivariate analysis were included in the multivariable logistic regression model to calculate adjusted odds ratios (aORs) with corresponding 95% confidence intervals (CIs). A p-value of $< 0.05$ was considered statistically significant. Multicollinearity of independent variables were assessed using variance inflation factor (VIF). Variables with $VIF < 5$ were considered to have no significant multicollinearity. Data were analyzed using STATA version14 statistical software (StataCorp LLC, College Station, TX, USA).

## Ethical considerations

This study was conducted as a public health response to investigate malaria related mortality among children under five at Mbale Regional Referral Hospital. Its primary intent was to generate evidence for immediate programmatic action to reduce deaths, rather than to produce generalizable knowledge. In agreement with the International Guidelines for Ethical Review of Epidemiological Studies by the Council for International Organizations of Medical Sciences (1991) and the Office of the Associate Director for Science, US CDC/Uganda, it was determined that this activity was not human subject research but public health practice (specifically, endemic disease control activity) and was therefore exempted from Institutional Review Board (IRB) review. This activity was reviewed by the US CDC and was conducted consistent with applicable federal law and CDC policy. §§See, e.g., 45 C.F.R. part 46, 21 C.F.R. part 56; 42 U.S.C. §241(d); 5 U.S.C. §552a; 44 U.S.C. §3501 et seq. Permission to access hospital records was granted by the Uganda Ministry of Health (MoH) and the Director of Mbale Regional Referral Hospital. All data were routinely collected, de-identified, and handled under the established confidentiality and data protection procedures.

## Results

### Socio-demographic and clinical characteristics of study participants

A total of 100 cases and 200 controls were enrolled in the study. Most children in both groups were aged <24 months; this proportion was higher among cases (73 [73%]) than controls (130 [65%]) (p = 0.006). The sex distribution was similar, with males comprising 61(61%) of cases and 106 (53%) of controls (p = 0.19).

Severe clinical manifestations were significantly more common among cases. Convulsions were reported in 62 (62%) of cases compared to 22 (11%) of controls, and cerebral malaria occurred in 42 (42%) versus 3 (1.5%), respectively (both p<0.0001). Severe anemia was present in 80 (80%) of cases compared to 99 (49.5%) of controls (p<0.0001), Gastrointestinal symptoms, including vomiting (73 [73%] vs. 96 [48%]; p<0.001) and diarrhea (41 [41%] vs. 50 [25%]), were more frequent among cases. Loss of consciousness was observed in 10 (10%) of cases compared to 3 (1.5%) of controls (p=0.001). Overall danger signs were present in 70 (70%) of cases compared to 46 (23%) of controls (p<0.0001). In contrast, hyperparasitemia, hypoglycemia, fever, jaundice, respiratory distress, number of severe symptoms, and duration of illness prior to presentation did not significantly differ between groups (Table 1).

Regarding healthcare access,care-seeking beyond 24 hours occurred more frequently among cases than controls (17 [17%] vs. 7 [3.5%]; p<0.0001), as did delays in hospital admission >4 hours (25 [25%] vs. 13 [6.5%]; p<0.001). Blood transfusion was administered to 62 (62%) of cases compared to 80 (41%) of controls (p=0.001). Length of hospital stay differed modestly between groups (p=0.04). No statistically significant differences were observed between cases and controls in distance to hospital, timing of admission (weekday versus weekend), admission shift, or time to blood transfusion (Table 1).

At multivariable analysis, several factors were significantly associated with increased odds of malaria-related mortality among under-five children with severe malaria. Convulsions on admission were strongly associated with higher odds of death (aOR = 17, 95% CI: 4.2–71), as was loss of consciousness (aOR = 14.0, 95% CI: 1.4–113). Children presenting with severe anaemia had over three times the odds of death compared to those without (aOR = 3.4, 95% CI: 1.4–8.2), while vomiting was also independently associated with increased mortality risk (aOR = 3.1, 95% CI: 1.4–6.9). Delays in seeking care significantly increased mortality risk - children who sought care more than 24 hours after symptom onset had markedly higher odds of death (aOR = 8.8, 95% CI: 2.3–34). Similarly, admission to hospital more than four hours after first contact with healthcare was associated with increased odds of death (aOR = 3.4, 95% CI: 1.3–9.2). Age categories, sex, diarrhea, blood transfusion, and overall danger signs were not significantly associated with mortality after adjustment (Table 2).

## Discussion

This study provides insights into the factors associated with malaria-related mortality among children under five years of age at a regional referral hospital in eastern Uganda. Factors significantly associated with death in this study were convulsions, loss of consciousness or coma, severe anemia, delays in healthcare seeking and vomiting.

Convulsions were strongly associated with malaria related death.

Children who presented with convulsions were 17 times more likely to die than those without convulsions. This finding is consistent with earlier studies which have reported convulsions having a strong association with malaria-related mortality [18–20]. Convulsions may signal severe cerebral involvement and should prompt urgent clinical intervention. Similarly, loss of consciousness or coma was associated with an eleven fold increased risk of death, consistent with studies from Rwanda and elsewhere highlighting altered consciousness is a critical prognostic indicator [8–10]. These neurological signs reflect underlying disease severity and and should prompt urgent clinical recognition and intervention.

Severe anemia was independently associated with mortality; the odds of dying were 3.4 times among children with severe anemia compared to those without. This finding is consistent with another study from Burkina Faso,which identified severe anemia as a major contributor to pediatric malaria mortality [21]. Severe anaemia in malaria results from rapid destruction of infected and uninfected red blood cells, bone marrow suppression, and splenic sequestration, leading to tissue hypoxia, metabolic acidosis, and potentially heart failure [22].Timely haemoglobin measurement and prompt access to blood transfusion are critical to reducing fatal outcomes [3].

The odds of death were 3.1 times higher among children who experienced vomiting, consistent with previous studies, such as one from Rwanda that linked vomiting with poorer outcomes in children under five years old [16]. Vomiting may

**Table 1. Socio-demographic and clinical characteristics of cases and controls among children under five admitted with severe malaria at Mbale Regional Referral Hospital, Uganda, 2020–2024.**

| Variables | Number of participants | | P-Value |
| --- | --- | --- | --- |
| | Cases (%) | Controls (%) | |
| **Age** | | | 0.006 |
| <24 | 73 (73) | 130 (65) | |
| 24-48 | 24 (24) | 48 (24) | |
| >48 | 3 (3) | 22 (11) | |
| **Sex** | | | 0.19 |
| Male | 61 (61) | 106 (53) | |
| Female | 39 (39) | 94 (47) | |
| **Convulsion** | | | **<0.0001** |
| Yes | 62 (62) | 22 (11) | |
| No | 38 (38) | 178 (89) | |
| **Hyper parasitemia** | | | 0.79 |
| Yes | 2 (2) | 5 (2.5) | |
| No | 98 (98) | 195 (97.5) | |
| **Cerebral Malaria** | | | **<0.0001** |
| Yes | 42 (42) | 3 (1.5) | |
| No | 58 (58) | 197 (98.5) | |
| **Diarrhea** | | | **0.004** |
| Yes | 41 (41) | 50 (25) | |
| No | 59 (59) | 150 (75) | |
| **Vomiting** | | | **<0.0001** |
| Yes | 73 (73) | 96 (48) | |
| No | 27 (27) | 104 (52) | |
| **Hypoglycemia** | | | **0.31** |
| Yes | 4 (4) | 4 (2) | |
| No | 96 (96) | 196 (98) | |
| **Severe anemia** | | | **<0.0001** |
| Yes | 80 (80) | 99 (49.5) | |
| No | 20 (20) | 101 (50.5) | |
| **Fever** | | | **0.47** |
| Yes | 98 (98) | 193 (96.5) | |
| No | 2 (2) | 7 (3.5) | |
| **Jaundice** | | | **0.47** |
| Yes | 2 (2) | 7 (3.5) | |
| No | 98 (98) | 193 (96.5) | |
| **Loss of consciousness** | | | **0.001** |
| Yes | 10 (10) | 3 (1.5) | |
| No | 90 (90) | 197(98.5) | |
| **Respiratory distress** | | | 0.18 |
| Yes | 7 (7) | 7 (3.5) | |
| No | 93 (93) | 193 (96.5) | |
| **Number of severe symptoms** | | | 0.89 |
| >3 symptoms | 11 (11) | 23 (11.5) | |
| < 3 symptoms | 89 (89) | 117 (88.5) | |
| **Number of days with the illness** | | | 0.64 |

*(Continued)*

**Table 1.** (Continued)

| Variables | Number of participants | | P-Value |
| --- | --- | --- | --- |
| | Cases (%) | Controls (%) | |
| > 4 days | 16 (16) | 28 (14) | |
| < 4 days | 84 (84) | 172 (86) | |
| **Days of hospital stay** | | | **0.04** |
| > 1 week | 82 (82) | 181 (90.5) | |
| < 1 week | 18 (18) | 19 (9.5) | |
| **Part of the week** | | | 0.67 |
| Week day | 19 (19) | 34 (17) | |
| Weekend | 81 (81) | 166 (83) | |
| **Blood transfusion** | | | **0.001** |
| Yes | 62 (62) | 80 (41) | |
| No | 38 (38) | 115 (59) | |
| **Shift on which the patient was admitted** | | | 0.06 |
| Morning | 52 (52) | 111 (55.5) | |
| Evening | 12 (12) | 40 (20) | |
| Night | 36 (36) | 49 (24.5) | |
| **Danger Signs** | | | **<0.0001** |
| Yes | 70 (70) | 46 (23) | |
| No | 30 (30) | 154 (77) | |
| **Time to health care** | | | **<0.0001** |
| <12hrs | 61 (61) | 164 (82) | |
| 12-24hrs | 22 (27) | 29 (14.5) | |
| >24hrs | 17 (17) | 7 (3.5) | |
| **Time to hospital admission** | | | **< 0.001** |
| 1hrs | 40 (40) | 88 (44) | |
| 1-4hrs | 35 (35) | 99 (49.5) | |
| >4hrs | 25 (25) | 13 (6.5) | |
| **Time before blood transfusion** | | | 0.9 |
| >3 hrs | 42 (65.6) | 53 (64.6) | |
| <3hrs | 22 (34.4) | 29 (35.4) | |
| **Distance to Hospital** | | | 0.17 |
| < 5Km | 25 (25) | 56 (28) | |
| 5-10Km | 36 (36) | 51 (25.5) | |
| >10Km | 39 (39) | 93 (46.5) | |

indicate more severe systemic illness or impaired oral drug absorption, which could delay effective treatment and worsen outcomes. Uncontrolled vomiting in severe malaria can contribute to hypovolemic shock and dehydration, which are potentially fatal if not promptly corrected [23, 24].

Delays in seeking care and hospital admission were critical contributors to mortality. Children who presented for care more than 24 hours after symptom onset were associated with higher odds of death. While hospital admission delays exceeding four hours after initial healthcare contact were associated with a near fourfold increase in mortality. These findings are consistent with other studies highlighting that delays in accessing timely care significantly elevate the odds of death [3, 16, 25]. It should be noted that delays in care and clinical severity are interrelated, and some of the observed association may reflect more severe disease at presentation.

**Table 2. Factors associated with mortality among under-five children with severe malaria, Mbale Regional Referral Hospital, Uganda 2020–2024.**

| Variables | Number of participants | | cOR (95%, CI | aOR (95%, CI |
|---|---|---|---|---|
| | Cases (%) | Controls (%) | | |
| **Age** | | | | |
| <24 | 73 (73) | 130 (65) | 4.1 (1.2-14) | 2.8 (0.9-13) |
| 24-48 | 24 (24) | 48 (24) | 3.6 (0.9-13) | 1.6 (0.31-8.3) |
| >48 | 3 (3) | 22 (11) | Ref | |
| **Sex** | | | | |
| Male | 61 (61) | 106 (53) | Ref | |
| Female | 39 (39) | 94 (47) | 0.72 (0.44-1.2) | 0.62 (0.30-1.2) |
| **Convulsion** | | | | |
| Yes | 62 (62) | 22 (11) | 13.2 (7.2-24) | **17 (4.2-71)** |
| No | 38 (38) | 178 (89) | Ref | |
| **Diarrhea** | | | | |
| Yes | 41 (41) | 50 (25) | 2.1 (1.3-3.5) | 1.7 (0.76-3.6) |
| No | 59 (59) | 150 (75) | Ref | |
| **Vomiting** | | | | |
| Yes | 73 (73) | 96 (48) | 2.9 (1.7-4.9) | **3.1 (1.4-6.9)** |
| No | 27 (27) | 104 (52) | Ref | |
| **Severe anaemia** | | | | |
| Yes | 80 (80) | 99 (50) | 4.1 (2.3-7.2) | **3.4 (1.4-8.2)** |
| No | 20 (20) | 101 (50) | Ref | |
| **Loss of consciousness** | | | | |
| Yes | 10 (10) | 3 (1) | 7.3 (1.9-27) | **14 (1.4-113)** |
| No | 90 (90) | 197 (99) | Ref | |
| **Blood transfusion** | | | | |
| Yes | 62 (62) | 80 (41) | Ref | |
| No | 38 (38) | 115 (59) | 2.3 (1.4-3.8) | 1.1 (0.46-2.5) |
| **Danger Signs** | | | | |
| Yes | 70 (70) | 46 (23) | 7.8 (4.5-13) | 0.98 (0.23-4.1) |
| No | 30 (30) | 154 (77) | Ref | |
| **Time to health care** | | | | |
| <12hrs | 61 (61) | 164 (82) | Ref | |
| 12-24hrs | 22 (27) | 29 (15) | 2 (1.1-3.8) | 2 (0.85-4.7) |
| >24hrs | 17 (17) | 7 (3.5) | 6.5 (2.6-16) | **8.8 (2.3-34)** |
| **Time to hospital admission** | | | | |
| 1hrs | 40 (40) | 88 (44) | Ref | |
| 1-4hrs | 35 (35) | 99 (50) | 0.8 (0.45-1.3) | 0.64 (0.29-1.4) |
| >4hrs | 25 (25) | 13 (6.0) | 4.2 (1.9-9.1) | **3.4 (1.3-9.2)** |

Ref: reference; cOR: crude odds ratios; aOR: adjusted odds ratios; CI: confidence interval.

Overall, these findings highlight the need for early recognition of neurological symptoms, prompt referral, and timely hospital admission to improve survival. Strengthening community health systems, improving referral pathways, and training frontline healthcare workers to identify danger signs could help reduce malaria-related child mortality. Targeted interventions in rural and hard-to-reach areas may also address the heightened risk associated with delayed care.

## Limitations

This study had some limitations: First, it was retrospective, relying on secondary data from hospital records, which may be subject to inaccuracies or incomplete documentation or recall bias. This is particularly relevant for variables such as symptom onset and time to care, which could be misclassified and potentially affect the observed magnitude of associations for delayed care-seeking. Second, the study was also conducted at a single referral hospital, which may not fully represent the broader population of children with severe malaria in Uganda or other regions. Furthermore, some adjusted estimates had wide confidence intervals, likely reflecting limited precision due to sparse data for certain clinical variables; these estimates should therefore be interpreted with caution. Despite these limitations, the study had notable strengths. It employed a well-defined unmatched case–control design with a sufficient sample size to detect meaningful associations. Controls were systematically sampled from a large pool of eligible patients, minimizing selection bias. Furthermore, the extended five-year study period allowed for a comprehensive assessment of severe malaria mortality trends in a high-burden, resource-limited setting, providing valuable evidence to inform targeted interventions.

## Conclusion

Malaria-related mortality among children under five at MRRH was independently associated with severe neurological signs (convulsions and loss of consciousness), severe anemia, vomiting, and delays in seeking care beyond 24 hours after symptom onset. These findings indicate that both severe disease at presentation and delayed access to treatment contribute substantially to pediatric malaria deaths. Reducing malaria-related mortality in this setting may require a dual strategy: strengthening early recognition and prompt management of severe malaria at hospital level through effective triage, continuous staff training, and reliable access to blood transfusion and antimalarials while simultaneously improving community awareness of danger signs and reinforcing referral systems, particularly through Village Health Teams, to reduce delays in care-seeking. Integrating both facility-based and community-level interventions could help achieve sustained reductions in malaria-related deaths among children under-five in similar high-burden settings.

## Supporting information

**S1 Data. Dataset supporting the findings of this study.**
(XLS)

## Acknowledgments

We extend our appreciation to Mbale Regional Referral Hospital administration for their overall coordination and leadership during the study and to the hospital data management team for the active participation. Lastly, we recognize the Uganda Public Health Fellowship Program for providing technical oversight, coordination, and funding throughout the study

## Author contributions

**Conceptualization:** Patrick Kwizera, Richard Migisha, Gerald Rukundo, Benon Kwesiga.

**Investigation:** Patrick Kwizera.

**Methodology:** Patrick Kwizera, Charity Mutesi, Gerald Rukundo, Steven Ndugwa Kabwama.

**Resources:** Alex Riolexus Ario.

**Supervision:** Benon Kwesiga.

**Writing – original draft:** Patrick Kwizera, Richard Migisha, Steven Ndugwa Kabwama.

**Writing – review & editing:** Patrick Kwizera, Richard Migisha, Charity Mutesi, Steven Ndugwa Kabwama, Lilian Bulage, Alex Riolexus Ario.

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
