## [Decision Letter · Decision Letter 0]

17 Dec 2025

PGPH-D-25-03409

Risk factors for malaria-related mortality among children under five at Mbale Regional Referral Hospital, Uganda, 2020–2024: a case-control study

Dear Kwizera,

Thank you for submitting your manuscript to PLOS Global Public Health. After careful consideration, we feel that it has merit but does not fully meet PLOS Global Public Health’s publication criteria as it currently stands. Therefore, we invite you to submit a revised version of the manuscript that addresses the points raised during the review process.

We look forward to receiving your revised manuscript.

Kind regards,

Collins Otieno Asweto, PhD

Academic Editor

Journal Requirements:

Additional Editor Comments (if provided):

Reviewers' comments:

Reviewer's Responses to Questions

**Comments to the Author**

1. Does this manuscript meet PLOS Global Public Health’s publication criteria? Is the manuscript technically sound, and do the data support the conclusions? The manuscript must describe methodologically and ethically rigorous research with conclusions that are appropriately drawn based on the data presented.? Is the manuscript technically sound, and do the data support the conclusions? The manuscript must describe methodologically and ethically rigorous research with conclusions that are appropriately drawn based on the data presented.

Reviewer #1: Partly

Reviewer #2: Yes

Reviewer #3: Yes

Reviewer #4: Yes

2. Has the statistical analysis been performed appropriately and rigorously?

Reviewer #1: No

Reviewer #2: Yes

Reviewer #3: Yes

Reviewer #4: Yes

3. Have the authors made all data underlying the findings in their manuscript fully available (please refer to the Data Availability Statement at the start of the manuscript PDF file)?

The PLOS Data policy requires authors to make all data underlying the findings described in their manuscript fully available without restriction, with rare exception. The data should be provided as part of the manuscript or its supporting information, or deposited to a public repository. For example, in addition to summary statistics, the data points behind means, medians and variance measures should be available. If there are restrictions on publicly sharing data—e.g. participant privacy or use of data from a third party—those must be specified.requires authors to make all data underlying the findings described in their manuscript fully available without restriction, with rare exception. The data should be provided as part of the manuscript or its supporting information, or deposited to a public repository. For example, in addition to summary statistics, the data points behind means, medians and variance measures should be available. If there are restrictions on publicly sharing data—e.g. participant privacy or use of data from a third party—those must be specified.

Reviewer #1: No

Reviewer #2: No

Reviewer #3: Yes

Reviewer #4: No

4. Is the manuscript presented in an intelligible fashion and written in standard English?

Reviewer #1: Yes

Reviewer #2: Yes

Reviewer #3: No

Reviewer #4: Yes

Reviewer #1: The author(s) states that they used secondary data that required no ethical review board, yet chooses not to attach it , at least for review process. I have attached the my comments in the document for further insights.

Reviewer #2: The study uses an unmatched 1:2 case–control design appropriate for evaluating mortality-related factors in severe malaria, with case and control definitions aligned with WHO criteria for reliability and comparability. Controls were systematically sampled from a large frame of over 32,000 survivors, reducing bias. Sample size was justified using an accepted approach, with 100 cases clearly explained. Data collection was robust, employing standardized tools, trained abstractors, and daily checks. Inclusion and exclusion criteria were explicit, and operational definitions for clinical variables were consistent. The multivariable logistic regression model was appropriate, including predictors significant at the bivariate level, with adjusted odds ratios and 95% CIs reported.

The associations like neurological signs, severe anemia, vomiting, and delayed care align with known predictors of severe malaria and death in low-resource settings. These findings support the importance of early recognition and timely care, consistent with cited literature.

Statistical analyses are appropriate. Descriptive statistics clearly characterize the groups. Chi-square tests are suited for comparing categorical variables. Logistic regression is correctly employed to control confounders.

• Several variables in Table 1 display inconsistencies or reversal of percentages (e.g., severe anemia appears very common in both cases and controls, but proportions differ between the text and the table). A thorough review of the table with clear labels is necessary.

• Some confidence intervals are very wide (e.g., aOR for loss of consciousness), reflecting limited precision and possibly sparse data, and this should be acknowledged.

• The manuscript should explicitly mention checks for multicollinearity and model fit

• Continuous variables were dichotomized; a rationale should be provided because categorization may reduce statistical power.

• Because delays in care and clinical severity are interrelated, discussion of possible confounding or mediation would strengthen the analysis.

The manuscript indicates that data were regularly collected, de-identified, and used with institutional approval. To comply with PLOS Global Public Health’s mandatory data availability policy, the authors are required to provide:

• A Data Availability Statement indicating where the dataset can be accessed publicly (repository link), or

• A formal explanation is provided when data cannot be shared, such as due to legal or ethical restrictions.

• The manuscript claims that the work was “not human subjects research” while also stating that “all experimental protocols were approved by the US CDC human subjects review board,” which is contradictory. The authors should clarify whether IRB review occurred or was waived due to the non-research classification.

No concerns about dual publication or research misconduct are evident.

The manuscript is generally well written, clear, and organized. It uses standard, understandable English. The sections follow a logical sequence, variables are defined, and interpretations are concise.

• Some typographical errors (spacing, punctuation) should be corrected.

• Proper table formatting is essential for aligning rows and values effectively.

• A few sentences in the Discussion reiterate points already made, tightening these would enhance readability.

Reviewer #3: The manuscript is insightful and very informative. However, the authors should please note the following and response as appropriate:

1. Kindly check the punctuation for line 178 to 180 and rectify accordingly. It is ambiguous.

2. Please, check and rewrite the results section for the socio-demographics and clinical characteristics (from line 178 to 191, especially), and make sure the format and figures are consistent and accurate. For example, the table presents 62(62%) cases and 22(11%) control with convulsion, but the table description says 42% vs. 10%. Very confusing, please where you got the 42% vs. 10% from. Similarly, with cerebral malaria, the table says 42(42%) cases and 3(1.5%) control but different in the description; just to give you a few.

3. Kindly make sure all the odds in the regression table are in the same decimal places to correspond with what is in the table description.

4. Kindly state the specific neurological symptoms identified in the study rather than the generalisation, as this statement could be misleading.

Reviewer #4: General comments:

The authors seek to determine the risk factors for malaria related mortality among children in Mbale regional referral Hospital. The strengths of this paper lies in its data analytical methods. Generally, it is a good paper but we need to find factors that do not cut across both in the cases and controls. These are the main factors that are causing mortality. For example, in severe anaemia, we have 80% dyeing and 50% surviving. What could have caused the 80% in the cases to die that was not in the survivors? This could bring out the gist of paper well. It could be a prevailing illness, NCD etc. Otherwise, the risk factors affecting the cases are the very factors affecting the controls in comparable measure in this paper.

Specific comments:

Abstract:

Line 28 - 30: is 300 out of 32,400 representative enough of the data?

Introduction:

Line 43 – 46, 56-60: A lot of plagiarism. Please reference peoples work. Consider the whole paper. There several sentences or paragraphs that you need to reference.

Problem statement:

Line 72 – 78: I want to believe that your problem statement is derived from that fact the mortality rate at MRRH is 2.7% way higher than the 0.6% on national average, What are the additional factors? Make mention of the additional factors that are actually causing mortality.

Methods:

The methods look fine but 300 representatives enough for the study? You could increase it to give you enough power for the study.

Line 116: In your sample size calculation, were you supposed to use the prevalence of severe anaemia or prevalence of severe malaria? Check that because this could affect your actual sample size.

Line 145 – 146: “Making accurate data extraction and analysis impossible outside” This sentence doesn’t make sense.

Results:

How do malaria cases relate with risk factors determined given thee catchment areas in the 16 districts?

Line 182 - 183: It’s the outcome death case categorizes the cases but before death they have exactly the same factors affecting them just as the controls (survivors), again what special factor kills the cases? If you consider the same number of participants for cases and controls, chances are high that there is no difference how the risk factors affect these 2 groups.

Line 199: You talk of multivariant analysis but I don’t see it the results? How does severe anaemia relate to say, convulsions, loss of consciousness and high parasitaemia for example?

Discussion:

Good discussion

Conclusion:

Line 280 – 281: State the results in the conclusion

**Do you want your identity to be public for this peer review?** For information about this choice, including consent withdrawal, please see our Privacy Policy..

Reviewer #1: **Yes:** Nicholas Siame AdamNicholas Siame AdamNicholas Siame AdamNicholas Siame Adam

Reviewer #2: No

Reviewer #3: No

Reviewer #4: **Yes:** Brian Asiimwe KagurusiBrian Asiimwe KagurusiBrian Asiimwe KagurusiBrian Asiimwe Kagurusi

---

## [Decision Letter · Decision Letter 1]

23 Feb 2026

PGPH-D-25-03409R1

Risk factors for malaria-related mortality among children under five at Mbale Regional Referral Hospital, Uganda, 2020–2024: a case-control study

Dear Dr. Kwizera,

Thank you for submitting your manuscript to PLOS Global Public Health. After careful consideration, we feel that it has merit but does not fully meet PLOS Global Public Health’s publication criteria as it currently stands. Therefore, we invite you to submit a revised version of the manuscript that addresses the points raised during the review process.

We look forward to receiving your revised manuscript.

Kind regards,

Helen Howard

Staff Editor

Journal Requirements:

Additional Editor Comments (if provided):

Reviewers' comments:

Reviewer's Responses to Questions

**Comments to the Author**

Reviewer #1: All comments have been addressed

Reviewer #2: (No Response)

Reviewer #3: All comments have been addressed

publication criteria? Is the manuscript technically sound, and do the data support the conclusions? The manuscript must describe methodologically and ethically rigorous research with conclusions that are appropriately drawn based on the data presented.? Is the manuscript technically sound, and do the data support the conclusions? The manuscript must describe methodologically and ethically rigorous research with conclusions that are appropriately drawn based on the data presented.

Reviewer #1: Yes

Reviewer #2: Yes

Reviewer #3: Yes

3. Has the statistical analysis been performed appropriately and rigorously?

Reviewer #1: Yes

Reviewer #2: Yes

Reviewer #3: Yes

4. Have the authors made all data underlying the findings in their manuscript fully available (please refer to the Data Availability Statement at the start of the manuscript PDF file)?

The PLOS Data policy requires authors to make all data underlying the findings described in their manuscript fully available without restriction, with rare exception. The data should be provided as part of the manuscript or its supporting information, or deposited to a public repository. For example, in addition to summary statistics, the data points behind means, medians and variance measures should be available. If there are restrictions on publicly sharing data—e.g. participant privacy or use of data from a third party—those must be specified.requires authors to make all data underlying the findings described in their manuscript fully available without restriction, with rare exception. The data should be provided as part of the manuscript or its supporting information, or deposited to a public repository. For example, in addition to summary statistics, the data points behind means, medians and variance measures should be available. If there are restrictions on publicly sharing data—e.g. participant privacy or use of data from a third party—those must be specified.

Reviewer #1: Yes

Reviewer #2: (No Response)

Reviewer #3: Yes

5. Is the manuscript presented in an intelligible fashion and written in standard English?

Reviewer #1: Yes

Reviewer #2: Yes

Reviewer #3: Yes

Reviewer #1: Authors have well addressed the comments I raised ealier on. It has been found that delayed care-seeking is a top risk factor; a recommendation for community-level education or strengthening the Village Health Team referral system would be highly relevant as a conclusion.

Reviewer #2: A part of the manuscript claims the activity was 'non-research,” but later mentions IRB approval, which seems contradictory, and these points need clarification. The authors should explicitly specify whether the study was ethically exempt, approved, or classified as public health surveillance, so they can address this inconsistency.

Lastly, some sentences in the manuscript lack citations to support the authority of certain concepts, so the authors should provide citations to support these statements.

Reviewer #3: The authors have made a very thorough revision, addressing the previous concerns very well. Few minor points needs to be revised to prevent potential misunderstanding by the readers:

1. Line 45 to 46, for clarify, kindly indicate where exactly the malaria cases and death increased, from 2019 to 2023.

2. Line 60 to line 63. Kindly check what you meant by what severe malaria is “Severe malaria is defined by WHO as clinical or laboratory evidence of vital organ dysfunction, with impaired consciousness/ coma, haemoglobin levels below g/dL, acute kidney injury, respiratory distress, circulatory collapse/shock, acidosis, jaundice, or disseminated intravascular coagulation”.

It is missing the key component of what malaria is, that is the presence of plasmodium in the blood, and not just the complication you have listed.

3. From line 185 to 189, kindly bring the percentage sign to the proportions in the description to distinguish between the absolute numbers and the proportions, just like your preceding format.

**Do you want your identity to be public for this peer review?** For information about this choice, including consent withdrawal, please see our Privacy Policy..

Reviewer #1: **Yes:** Nicholas Siame AdamNicholas Siame AdamNicholas Siame AdamNicholas Siame Adam

Reviewer #2: No

Reviewer #3: No

---

## [Decision Letter · Decision Letter 2]

22 Mar 2026

PGPH-D-25-03409R2

Risk factors for malaria-related mortality among children under five at Mbale Regional Referral Hospital, Uganda, 2020–2024: a case-control study

Dear Dr. Kwizera,

Thank you for submitting your manuscript to PLOS Global Public Health. After careful consideration, we feel that it has merit but does not fully meet PLOS Global Public Health’s publication criteria as it currently stands. Therefore, we invite you to submit a revised version of the manuscript that addresses the points raised during the review process.

We look forward to receiving your revised manuscript.

Kind regards,

Helen Howard

Staff Editor

**Journal Requirements:**

**Additional Editor Comments (if provided):**

Reviewers' comments:

Reviewer's Responses to Questions

**Comments to the Author**

Reviewer #2: (No Response)

Reviewer #3: (No Response)

publication criteria? Is the manuscript technically sound, and do the data support the conclusions? The manuscript must describe methodologically and ethically rigorous research with conclusions that are appropriately drawn based on the data presented.? Is the manuscript technically sound, and do the data support the conclusions? The manuscript must describe methodologically and ethically rigorous research with conclusions that are appropriately drawn based on the data presented.

Reviewer #2: Yes

Reviewer #3: Yes

3. Has the statistical analysis been performed appropriately and rigorously?

Reviewer #2: Yes

Reviewer #3: Yes

4. Have the authors made all data underlying the findings in their manuscript fully available (please refer to the Data Availability Statement at the start of the manuscript PDF file)?

The PLOS Data policy requires authors to make all data underlying the findings described in their manuscript fully available without restriction, with rare exception. The data should be provided as part of the manuscript or its supporting information, or deposited to a public repository. For example, in addition to summary statistics, the data points behind means, medians and variance measures should be available. If there are restrictions on publicly sharing data—e.g. participant privacy or use of data from a third party—those must be specified.requires authors to make all data underlying the findings described in their manuscript fully available without restriction, with rare exception. The data should be provided as part of the manuscript or its supporting information, or deposited to a public repository. For example, in addition to summary statistics, the data points behind means, medians and variance measures should be available. If there are restrictions on publicly sharing data—e.g. participant privacy or use of data from a third party—those must be specified.

Reviewer #2: Yes

Reviewer #3: Yes

5. Is the manuscript presented in an intelligible fashion and written in standard English?

Reviewer #2: Yes

Reviewer #3: Yes

**Reviewer #2:** Line 177--A part of the manuscript claims the activity was 'non-research”. The authors should clarify what that means. Line 177--A part of the manuscript claims the activity was 'non-research”. The authors should clarify what that means. Line 177--A part of the manuscript claims the activity was 'non-research”. The authors should clarify what that means. Line 177--A part of the manuscript claims the activity was 'non-research”. The authors should clarify what that means.

**Reviewer #3:** While the efforts to standardise the formatting of the sociodemographic data are noted, the current revision has introduced a significant reporting error. By removing the absolute numbers (n) and reporting only percentages, you have obscured the raw data and the denominator for each category.While the efforts to standardise the formatting of the sociodemographic data are noted, the current revision has introduced a significant reporting error. By removing the absolute numbers (n) and reporting only percentages, you have obscured the raw data and the denominator for each category.While the efforts to standardise the formatting of the sociodemographic data are noted, the current revision has introduced a significant reporting error. By removing the absolute numbers (n) and reporting only percentages, you have obscured the raw data and the denominator for each category.While the efforts to standardise the formatting of the sociodemographic data are noted, the current revision has introduced a significant reporting error. By removing the absolute numbers (n) and reporting only percentages, you have obscured the raw data and the denominator for each category.

Kindly revise all sociodemographic descriptions in the text and tables to the standard n (%) format (e.g., '130 (65%); 62 (62%)' and not ‘62(62)’ nor ‘62%’). Please, ensure this format is applied consistently throughout the entire manuscript, where applicable, to differentiate clearly between absolute counts and proportions.

**Do you want your identity to be public for this peer review?** For information about this choice, including consent withdrawal, please see our Privacy Policy..

Reviewer #2: No

Reviewer #3: No

**Figure Resubmissions:**While revising your submission, we strongly recommend that you use PLOS’s NAAS tool (https://ngplosjournals.pagemajik.ai/artanalysis) to test your figure files. NAAS can convert your figure files to the TIFF file type and meet basic requirements (such as print size, resolution), or provide you with a report on issues that do not meet our requirements and that NAAS cannot fix.

---

## [Editor Report · Decision Letter 3]

31 Mar 2026

Risk factors for malaria-related mortality among children under five at Mbale Regional Referral Hospital, Uganda, 2020–2024: a case-control study

PGPH-D-25-03409R3

Dear Epidemiologist Kwizera,

We are pleased to inform you that your manuscript 'Risk factors for malaria-related mortality among children under five at Mbale Regional Referral Hospital, Uganda, 2020–2024: a case-control study' has been provisionally accepted for publication in PLOS Global Public Health.

Best regards,

Julia Robinson

Executive Editor